# Characterization of Bunch Compactness in a Diverse Collection of *Vitis vinifera* L. Genotypes Enriched in Table Grape Cultivars Reveals New Candidate Genes Associated with Berry Number

**DOI:** 10.3390/plants14091308

**Published:** 2025-04-26

**Authors:** Marco Meneses, Claudia Muñoz-Espinoza, Sofía Reyes-Impellizzeri, Erika Salazar, Claudio Meneses, Katja Herzog, Patricio Hinrichsen

**Affiliations:** 1Instituto de Investigaciones Agropecuarias, INIA La Platina, Santiago 8831314, Chile; 4mmeneses@gmail.com (M.M.); esalazar@inia.cl (E.S.); 2Agronomy Faculty, Universidad de Concepción, Chillán 3780000, Chile; 3Centro de Biotecnología Vegetal, Facultad de Ciencias de la Vida, Universidad Andrés Bello, Santiago 8340008, Chile; sofia.rimpel@gmail.com; 4Agronomy Faculty, Pontificia Universidad Católica de Chile, Santiago 8331150, Chile; claudio.meneses@uc.cl; 5Julius Kühn-Institut, Institute for Grapevine Breeding, Geilweilerhof, 76833 Siebeldingen, Germany; katja.herzog@julius-kuehn.de

**Keywords:** bunch compactness, GBS, automated phenotyping, GWAS, grapevine, table grape

## Abstract

Bunch compactness (BC) is a complex, multi-trait characteristic that has been studied mostly in the context of wine grapes, with table grapes being scarcely considered. As these groups have marked phenotypic and genetic differences, including BC, the study of this trait is reported here using a genetically diverse collection of 116 *Vitis vinifera* L. cultivars and lines enriched for table grapes over two seasons. For this, 3D scanning-based morphological data were combined with ground measurements of 14 BC-related traits, observing high correlations among both approaches (R^2^ > 0.90–0.97). The multivariate analysis suggests that the attributes ‘berries per bunch’, ‘berry weight and width’, and ‘bunch weight and length’ could be considered as the main descriptors for BC, optimizing evaluation times. Then, GWASs based on a set of 70,335 SNPs revealed that GBS analysis in this same population enabled the detection of several SNPs associated with different sub-traits, with a locus for ‘berries per bunch’ in chromosome (chr) 18 being the most prominent. Enrichment analysis of significant and frequent SNPs found simultaneously in several traits and seasons revealed the over-representation of discrete functions such as alpha-linolenic acid metabolism and glycan degradation. In summary, the utility of 3D automated phenotyping was validated for table grape backgrounds, and new SNPs and candidate genes associated with the BC trait were detected. The latter could eventually become a selection tool for grapevine breeding programs.

## 1. Introduction

Grapevine (*Vitis vinifera* L.) is the most important Mediterranean fruit crop, in terms of both value and cultivated area. It has a rich genetic diversity evidenced by over 15,000 recognized cultivars, which are destined for different purposes such as wine production and/or fresh table grape consumption [1]. The rich genetic diversity of this species is directly linked to the development of early human settlements, with evidence strongly suggesting a dual domestication process in Eurasia that started 11,000 years BP [2]. This heritage now constitutes part of human culture, and the vast genetic resources of *V. vinifera* are the main substrate from which new cultivars are bred, with the aim of satisfying modern needs regarding yield and overall fruit quality [3], pest and disease resistance [4], adaptation to climate-change scenarios [5], and oenological characteristics [6], among many other factors.

Wine and table grapes conform to two separate genetic pools, explained to some extent by a bimodal domestication history and the prevalent use of each type [7,8]. These genetic pools may be associated with the differences observed in certain characteristics such as berry size and shape [9], as well as bunch length and compactness [10]. For example, the berries of wine cultivars tend to be smaller as many chemicals associated with the character of wine are concentrated in the berry skin; this is in contrast to table grape berries, which are preferably as large as possible for fresh consumption [11]. In the case of the seeds, these organs are the main source of tannins, a group of complex aromatic compounds that enrich wine; in contrast, seeds are undesirable for most consumers of fresh table grapes [12,13]. Furthermore, the breeding dynamics between both classes also differ as the wine industry, which has relied mostly on renowned cultivars, has experienced low varietal turnover over decades or even centuries [14]. Recent trends, however, indicate positive interest in incorporating genetic changes through breeding towards more sustainable viticulture in the face of the challenges posed by climate change [5,6]. In this sense, incorporating disease resistance could reduce the use of pesticides, which is necessary for the sustainability of both the table [15] and wine grape industries [6]. Moreover, additional traits related to mechanical resistance could also contribute to the prevention of diseases by selecting the optimal epicuticular waxes [16] or less compact bunches [17,18].

Bunch compactness (BC) is a complex trait that has been defined as “a balance between rachis extent and berries total volume” or the portion of the morphological volume of the bunch that is not filled by berries, with both factors depending on genetic determinants and environmental conditions [19]. For several reasons, understanding the genetics, biology, and agronomical performance of this trait may hold promise. First, the reduced ventilation and the difficulty for pesticides to properly cover berries in tighter bunches are closely correlated with susceptibility to fungal diseases [20,21,22,23]. Another reason for selecting varieties with reduced BC is to diminish the incidence of resistance to new molecules, which is quite frequent in pathogens such as *Botrytis cinerea* [24]. In addition, a loose bunch is associated with better color development due to direct sunlight exposure [25]. Light also determines parameters such as juice content, flesh pH, sugars, organic acids content, and concentration of secondary metabolites [26]. Specifically in table grapes, the pruning of clusters is required to obtain the optimal bunch shape, which is a prerequisite for a marketable product [27]. It is well known that plant growth regulators such as gibberellic acid (GA_3_) are applied to modify cluster architecture [28]. However, the outcomes of such treatments differ depending on the varietal response, application timing, dosage, and environmental conditions [29]. More importantly, reducing the use of chemicals is expected to become the norm, due to their impacts on the environment and their low popularity among consumers.

However, the measurement of BC is quite challenging when contrasted with the relative simplicity of evaluating most sub-traits [19]. For instance, scoring lengths of peduncle and other rachis sub-structures is simple but laborious [30], requiring the measurement of berry volume and weight [31,32,33]. Briefly, two components mainly describe BC: cluster morphological volume, which depends on various architectural aspects; and the solid component, determined by berry number and individual volume [19]. Recent methodologies that have been introduced to estimate and quantify some of these characteristics have mostly focused on 3D scanning systems. However, most of these works have been carried out on wine cultivars, with very few cases focused on table grapes. Interestingly, one of the first papers describing the basics of bunch architecture compared wine varieties (‘Riesling’ and ‘Chardonnay’) versus table grapes (‘Sultanina’ and ‘Exotic’), harboring compact versus loose bunches, respectively [34].

In recent years, tridimensional scanning platforms have been implemented to explore the spatial characteristics of different plant species. The main focus of the first work on grapevines was to reconstruct 3D models in a semi-automated way [35]; more recently, convolutional neural network algorithms have been applied for image recognition and automated data extraction [17,36,37]. These procedures have been shown to be highly consistent with the direct measurements of berry and rachis sub-structures. They can proceed in a non-invasive way, which is significant when samples are not abundant, as in the case of working with germplasm repositories.

The genetics underlying BC remain to be further characterized. However, some genes have already been proposed in an attempt to explain differences in particular traits, such as *VvUCC1* for pedicel and ramification length [38] and *VvGRF4* for pedicel and rachis length [39]. Meanwhile, quantitative trait loci (QTLs) have also been proposed. For instance, Marguerit et al. [40] found two QTLs for peduncle length and three QTLs for rachis length. Correa et al. [30] identified 19 QTLs for peduncle, rachis, lateral shoulder, and pedicel lengths and node number along the main axis in a biparental table grape population (‘Ruby Seedless’ x ‘Sultanina’). Based on a crossing of ‘Calardis Musqué’ x ‘Villard Blanc’ (n = 149), Richter et al. [41] proposed the existence of 30 QTLs associated mainly with shoulder, rachis, and pedicel lengths. Using a population with a complex pedigree of American Vitis species (n = 112), Underhill et al. [42] found only two stable QTLs reproduced in two consecutive years for berry-related traits but not a single stable QTL associated with rachis architecture or bunch compactness, suggesting that these latter traits are subject to strong environmental influence. Later, through association mapping based on a panel of over 100 cultivars used as wine varieties, a group of 27 SNPs belonging to 10 different genes specifically associated with the length of the first ramification of the rachis was identified [43]. These same authors described a discrete number of genes associated with BC through sequencing and SNP detection on 183 candidate genes.

Complex traits such as bunch compactness need to be assessed by combining the study of diverse populations and precise, powerful analytical methods, with the purpose of expanding the discovery of new genetic determinants from which new proposals could arise. Considering this, our main objective in this work was to characterize the sub-traits involved in the expression of the BC phenotype; we used data obtained from a diverse collection of varieties used as table grapes via automated phenotyping from 3D scans and conventional ground measurements. The phenotypic data were coupled to the genotyping data generated by genotyping-by-sequencing (GBS) to analyze (through genome-wide association studies; GWASs) the genetic determinism of this trait, identifying the loci and closest genes involved in each case.

## 2. Results

### 2.1. Relationships Among the Traits Determining Bunch Compactness

A total of 14 traits were characterized using standard phenotyping methods (ground measurements), and eight traits were estimated using automated phenotyping based on 3D scans (Figure 1).

The traits described in Figure 1 were characterized in a collection of *V. vinifera* L. cultivars composed of 116 genotypes, aiming to maximize the diversity of BC-related features. The collection contained varieties from diverse genetic backgrounds and provenances that were classified according to their utilization as table grapes, wine, or double-purpose varieties (Appendix A).

When data from the ground measurements across two seasons were examined using multivariate analysis, several relationships among the traits and individuals were suggested. As is shown in Figure 2, a total of four trait groups were suggested via the principal component analysis:The first group shows close association between berries per bunch and number of lateral branches;The second group is related to the structural component of BC. This group includes the traits bunch weight, stalk length, stalk weight, and first and second branch lengths;The third group is related to the solid component of BC associated with berry characteristics, including length, width, and fresh weight. Interestingly, it also includes pedicel length, which appears to be more related to the solid component of the bunch rather than the structural one;Finally, the fourth group is composed mainly of seed-related traits, showing variable behavior among seasons and inversely correlated with other traits, such as the number of berries per bunch during the second season.

Biplots of the principal component analysis for seasons 1 and 2 explained 66.9% and 65% of the total phenotypic variance, respectively. The contribution to the first component was higher for two groups (2 and 3; structural and solid components, respectively). Meanwhile, the other two (1 and 4) contributed largely to the second component, with more variability observed among seasons (Appendix A). In more detail, of the traits studied to describe BC, berry fresh weight, berry width, bunch weight, and stalk length presented a high contribution to the first principal component (Appendix A). Meanwhile, berries per bunch and seed-related traits contributed largely to the second principal component (Appendix A). As for the relationships of the varieties themselves, their attributes presented some differentiation according to their usage, as suggested by the biplot in Figure 2.

Other relationships among traits to describe the overall compactness of the bunches were proposed in the work by Tello and Ibáñez [18], according to which several quantitative compactness indexes (CIs) can be calculated from different combinations of attributes. In this work, a total of 12 of those indices, namely CI-01, CI-02, CI-03, CI-06, CI-08, CI-10, CI-12, CI-14, CI-16, CI-17, CI-18, and CI-19, were calculated. The indices were inspected to verify data distributions. As these were composite variables, they exhibited different ranges and values, summarized in Appendix A. Interestingly, the indexes CI-01, CI-02, CI-03, CI-06, CI-08, CI-10, CI-12, and CI-14 showed different magnitudes but similar distributions, as can be seen in Appendix A. CI-16 was the only index that showed negative values; the remaining indexes, namely CI-17, CI-18, and CI-19, showed strikingly different distributions, highly skewed to the left in Appendix A (loose bunches) with some extreme values to the right (compact bunches).

Overall, the multivariate analysis and examination of the CI suggest that a few traits could succinctly reflect the complexity of BC, namely, berries per bunch, berry fresh weight and width, bunch weight, and stalk length. Regarding CIs, CI-03 is an index that is relatively easy to measure based on the availability of 3D scan technologies, which are addressed in the following section.

### 2.2. Automated and Standard Phenotyping Showed High Correlation Values

The automated phenotyping method to characterize traits describing BC proposed by Rist et al. [17] was also implemented according to the phenotyping scheme (Figure 1). When the dataset also incorporated the variables inferred by the 3D Bunchtool software v2.0, the same principal component analysis method suggested similar relationships as expected for traits with direct correlations, such as the number of berries per bunch and others of a similar nature (Appendix A). Overall, the correlation matrix among all variables showed a high degree of correlation based on the R^2^ values of all inferred traits with their ground measurement counterparts for both seasons, as seen in Figure 3.

One key aspect of integrating automated phenotyping is the possibility of replacing time-consuming measurements, which in turn could accelerate and even expand data acquisition. In fact, the correlation value of the number of berries, which is one such laborious evaluation, between both methods showed an R^2^ of 0.96 and 0.97 for the first and second seasons, respectively. Other pairs of closely related traits, such as convex hull volume versus bunch weight, grape length versus stalk length, and total volume versus bunch weight, showed average R^2^ values of 0.94, 0.93, and 0.91, respectively (Figure 4).

Overall, a few negative correlations among variables were detected in the correlation matrix analysis. Only seed-related traits, namely seed number and seed fresh weight, showed negative R^2^ values, which were relatively low in magnitude for all the matrices; for example, the maximum negative R^2^ values reached a value of ~0.5 when comparing seed fresh weight with berries per bunch in the second season; meanwhile, all other comparisons were below 0.30 (Figure 3b).

### 2.3. Genetic Diversity and Structure of the Collection of Samples

To obtain a representative genetic characterization of the collection of varieties and segregating lines, a GBS approach was taken. The discovery of variants was performed using the reference genome PN40024 of *V. vinifera* L. [44]. A total of 685,934 variants were discovered based on a minimum criterion for filtering (minimum quality: Q20). These data were inspected to define further criteria for quality control and informativeness, evaluating parameters such as read depth, mapping quality, and marker density, among others (Appendix A). In this context, further filtering using quality parameters (see Methods) gave a set of 114,515 potentially informative variants, representing 36,838 InDels, 7342 tri-allelic SNPs, and 70,335 bi-allelic SNPs. However, only biallelic SNPs were considered when performing further downstream analyses; their distribution along the reference genome is shown in Appendix A. Based on hierarchical methods performed using genetic data from this variety set, a highly structured (K = 4) population was determined, and three groups associated with usage (table, wine, and double-purpose) and further defined by their provenance/origin were identified (Figure 5 and Appendix A). Interestingly, the 30 table grape varieties included in the study, belonging to five different crossings, were distributed in two neighboring clades of table grape groups, mainly composed of Californian cultivars, coinciding with parentals.

### 2.4. Genotype–Phenotype Association Studies

Given the population structure of the variety set (see Figure 5 and Appendix A) and the existing relatedness matrix among the different cultivars, association studies based on mixed linear models that can control for both factors were performed [46]. Furthermore, a correction for multiple hypothesis testing was applied following the false discovery ratio method [47] using the standard threshold for significance (FDR < 0.05). Regarding the phenotypic data, as was stated for principal component analysis (Section 2.1), based on multivariate analyses, a discrete number of traits were able to represent the overall variability of the BC phenotype in the collection. One such attribute corresponded to the number of berries per bunch (BB), showing one of the strongest signals located on chr 18, and this was identified in both evaluated seasons (Table 1). Moreover, other sites for BB were sparsely detected during the second season on chr 2, 4, 5, 6, 14, and 17 (Figure 6).

Regarding other traits describing BC, such as bunch weight and stalk length (corresponding to the structural component) and berry fresh weight and length (corresponding to the solid component), a discrete number of SNPs showing association were found, some of them in a single season. In the case of seed fresh weight, a strong signal on chr 18 was identified, encompassing several SNPs, coincidentally with the main QTL reported for seedlessness [48,49]. In fact, from a total of 111 and 104 SNPs found for this trait, chr 18 alone represented 90 and 71 SNPs for the first and second seasons, respectively.

In relation to the analysis of compactness indexes (CIs) as input for quantitative values in association studies, we found associated SNPs for nine CIs. For ‘CI-03’ (a convenient index for us since it can be fully measured with 3D Bunchtool), association signals were identified on chr 18 and 14 (second season) (Appendix A). All the significant SNPs were annotated for their potential effects on the closest genes in a window of 5 Kb (2.5 Kb upstream/downstream) of the position site (Table 1). The full list of significant SNPs detected for the remaining traits and compactness indexes is presented in Appendix A.

### 2.5. Survey of the Closest Genes Around the Most Frequent SNPs Showing Association with the Traits Describing BC

In total, 639 significant signals for 21 variables were found, with 12 corresponding to BC-related traits and 9 corresponding to CIs characterized over two consecutive seasons (season 1: 281; season 2: 358; Appendix A). Interestingly, several SNPs were found to be significantly associated with multiple traits. Considering this, a total of 221 and 194 unique sites were identified for the first and second seasons, respectively, and 92 of them (28%) appeared in both seasons (see Venn diagram in Appendix A). These 92 common SNPs were annotated as previously described; the predicted effects for the complete dataset are summarized in Appendix A. Furthermore, this gene set was used for enrichment analysis to detect over-represented gene ontology (GO) terms. In summary, 64 of the 92 genes were mapped, and the analysis detected significant enrichment in the functions detailed in Appendix A. The most relevant GO categories were glycan degradation (VITVI18G02248) and alpha-linolenic acid metabolism (Vitvi18g02142 and Vitvi18g02139), which are involved in jasmonate metabolism.

Furthermore, to dissect the possible functional implications based on the gathered data, genes that were involved in signal associations with multiple traits were also inspected, even when sharing the SNP in the same position. Significant SNPs detected in SFW were filtered to reduce signal noise, given the high density of sites detected in chr 18. The list is presented in Table 2, which also includes genes previously suggested in alpha-linolenic acid metabolism, such as Vitvi18g02139, Vitvi18g02142, and Vitvi18g03160.

## 3. Discussion

### 3.1. Relevance of Bunch Compactness in Viticulture

Despite its importance for disease resistance and fruit quality, bunch compactness has been scarcely studied due to its polygenic control and the inherent difficulties in gauging it. In addition, as both environmental conditions and vineyard management highly influence the phenotype, achieving reproducibility can be difficult when data are collected over different seasons (revised by [41]). In this regard, the study of genetic determinism of such traits could be challenging. However, the high heritability described for several rachis architectures and estimated BC-related traits [30] reassures the relevance of genetic factors as determinants for the expression of these traits. In fact, efforts to elucidate the genetics behind bunch compactness have shown interesting results over the past few years [38,39,50], a search that has been carried out essentially against a wine-varietal background. Therefore, studying BC in table grapes using the current state-of-the-art analytical tools will help to elucidate which genetic and phenotypic factors are determinants of the expression of compactness.

### 3.2. Major Findings

As previously stated, the main objective was to address BC variability, extending the current theoretical and methodological approaches to a collection of grapevines with high genetic diversity enriched in table grapes. In summary, three main findings contributed to the understanding of the subject of study:The identification of sub-traits that are the main determining factors describing BC;The confirmation of automated phenotyping as a robust methodology to study BC in table grapes;The identification of novel genetic variants (SNPs) associated with the BC-related traits.

Each one of these findings is discussed in the following sections.

#### 3.2.1. Which Traits Are the Main Determinants of Bunch Compactness?

According to the literature, [19] two factors determine BC: the solid component and the structural component. Our data support this theory, identifying participant traits such as berry fresh weight and berry length and width for the former and bunch weight and bunch/stalk length for the latter. However, other important aspects that determine BC could be the number of berries per bunch and the number of lateral branches (with both being closely related to each other). These two attributes could be more sensitive to environmental effects in comparison to solid or structural-related traits. They could account for the variability in BC in terms of major seasonal effects, as suggested by our multivariate analyses (Figure 2). In accordance with this, previous studies identifying QTLs for total berry number have reported a lack of recurring regions associated with the trait among seasons, which could reflect the inherent variability of this attribute [51]. Even in some cases that consider several BC-related traits simultaneously, researchers have indicated that berries per bunch contributed the most to the overall variability observed among seasons and populations [50].

The dataset in this study was enriched for table grapes. In this regard, the notion that wine and table grapes constitute two quite different gene pools has been well-described [2,7,10]. The genetic differences between both groups have a phenotypic correspondence, such as for berry size; this is preferred to be as large as possible in table grapes, whereas there is a preference for rather small berry sizes in wine varieties, where the largest possible content of skin-associated pigments and aromatic compounds is highly valued. Interestingly, a considerable fraction of the double-purpose varieties included in this work overlap with wine varieties (see Figure 2; principal component analysis of phenotype data), a pattern also partially evidenced in the hierarchical clustering based on genetic data (Figure 5); most probably, this overlap is because such categorization is based on the registered uses of each genotype which could not necessarily reflect the most prominent use in each case: e.g., many old wine varieties (such as Listán Prieto) have been also used for fresh consumption, and consequently they had been registered as double-purpose varieties. Both the BC-related phenotypic data and the genetic clustering evidenced in this work (the latter based on over 70,000 SNPs covering the whole grapevine genome) could eventually be used to reclassify some of the varieties included in this work as being closer to the wine or table grape groups.

Cluster size also tends to be divergent between the two groups, following the same tendency as berry size. Our correlation matrix of BC-related attributes showed a scarcity of negatively correlated variables; this could suggest that the traits selected in table grapes have been positively selected to simultaneously enhance berry size and cluster size. Regarding negative correlations, only seed-related attributes suggested mild negative R^2^ values regarding the rest of the dataset, which were variable among seasons (Figure 3). Seedlessness is a key trait sought in modern table grape cultivars, and it can have a diminishing effect on both berry and bunch weight, as has been comprehensively studied by Costantini et al. [52], even when comparing individuals that are somatic variants of the characteristic.

#### 3.2.2. Automated Phenotyping Based on 3D Scans Is a Highly Reliable Methodology to Address the Complexity of Traits Determining BC

The present work validates the implementation of automated phenotyping based on 3D scans to study table grapes. This is critical information when considering that overall bunch and berry dimensions in table grapes can reach extreme values when compared to wine grapes. This is supported by the correlation values obtained (Figure 3 and Figure 4) being close to those reported in [17] for several traits. These results suggest positive prospects if the phenotyping is fully automated; the high precision of this method and the fast acquisition of the data can relieve the phenotyping bottleneck that occurs when data are gathered using traditional measurements, as it is a high-throughput method suitable for the kinds of large populations expected under a conventional breeding scheme. This has already been observed in several breeding populations derived from crossing wine varieties—for example, ‘Riesling x Sauvignon Blanc’, ‘Calardis Musque’ x ‘Villard Blanc’, and ‘Dakapo‘ x ‘Cabernet Sauvignon’ [50]—however, this is the first report on table grapes, which have considerably larger bunches. In fact, powering up the number of observations of bunches per genotype can help estimate the true value of the population more precisely as BC is a complex trait, and its variability can vary greatly within a small sample size.

Although some small caveats, such as high variability in terms of berry shape—which could deviate from the program’s inference (since it adjusts spherical shapes by default)—could arise regarding table grapes, there was no considerable deviation. The advantages of the reproducibility and standardization of the measurements can represent an outstanding opportunity to address BC in divergent genetic backgrounds and compare different populations, seasons, and locations.

#### 3.2.3. How Different Are the Loci and Genes Found in Different Populations?

The association studies led to the identification of a total of 639 SNPs across the different seasons and attributes explored. Of all the studied traits, strong signals were detected regarding the number of berries per bunch, concentrated in chr 18, as observed in [50], with other SNPs distributed in chr 2, 4, 5, 14, and 17. However, when the physical positions of previously reported QTLs associated with berry number (BN; i.e., berries per bunch in the present work) are compared with the SNPs reported in this study, the peaks are distributed in contrasting regions. Specifically, flanking markers VMC2A3 (~0.95 Mb) and VMC8F4.2 (~10 Mb) contain QTLs surpassing the threshold LOD value for three different populations across several seasons. Meanwhile, the significant SNPs located in chr18 for the same trait found here spanned from 27.8 Mb to 32.8 Mb.

The use of different mapping populations to uncover genomic regions and loci associated with BC has not necessarily rendered consistent results. This has at least two different points of view: the genetic architecture of the traits and the genomic regions found. For instance, Correa et al. [30] found 19 QTLs associated with different sub-traits, all related to rachis architecture (number of nodes or branches and length of rachis, peduncle, shoulder, and pedicel), distributed on six linkage groups (chr 5, 8, 9, 14, 17, and 18), among which LG-5 and LG-18 harbored the largest number of QTLs. That study was carried out on a biparental crossing (‘Ruby Seedless’ x ‘Sultanina’) and, therefore, was limited to the allelic diversity present in parents [53]; therefore, the application of their results could eventually be suitable to lines with similar genetic backgrounds. A similar number of QTLs (24) was later identified in a crossing using wine varieties (‘Calardis Musque’ ×’Villard Blanc’) [41]. In this case, QTLs were also linked to architectural traits, including stalk (rachis), shoulder, and pedicel length. Still, they were distributed in different positions with respect to the table grape crossing, located in eight genomic clusters (chr 1, 2, 3, 10, 12, 17, and 18). Even the QTLs found on LG17 corresponded to different traits and genomic segments on this chromosome. When the same group expanded the genetic background, including the three aforementioned crossings derived from wine varieties (‘Calardis Musque’ x ‘Villard Blanc’; ‘Riesling’ x ‘Sauvignon Blanc’; and ‘Dakapo’ x ‘Cabernet Sauvignon’) [50], they found QTLs for bunch length on chr 1, 2, 4, 8, 9, and 15, but also for volume components, e.g., berry number and volume (chr 4, 6, 8, 10, 12, 17, and 18). As can be appreciated, just a few genomic regions were coincident, notably on chr 8, 17, and 18. Compared to the present work, there are some concordances for traits and the genomic regions associated, in particular, for the number of berries per bunch (BB in this work); a strong signal was found for this trait in a region of approximately 3 Mb in the extremity of chr 18 (two seasons), plus two loci at chr 4 and 17 but only in one season. Therefore, this approach presents a substantial difference from the previous works based on linkage maps built on controlled crosses, which contained a reduced number of recombination events in the progenies [53].

### 3.3. Other Aspects Worthy of Consideration

#### 3.3.1. Phenotyping Efficiency Could Be a Key Aspect with Which to Study BC and Achieve Data Reproducibility

Despite its relevance for productive reasons, the first studies aimed at identifying the main components of bunch compactness (BC) and rachis architecture (RA) were quite recent. Shavrukov et al. [34] found that inflorescence (rachis) length and inter-arms distance were the most relevant traits determining the compactness level. This was later confirmed by Correa et al. [30], who identified seven sub-traits as the most relevant, with rachis length, coincidentally, being one of them. Still, lateral shoulder length and node number were also among the most informative. When a multivariate analysis was conducted, considering the 14 sub-traits studied using the ground measurements, they grouped into four well-separated vectors, as shown in Figure 2, obtaining very similar patterns in both seasons. This is relevant because the notion of measuring fewer parameters (for example, one per group of traits) and obtaining the same result could be proposed, which must be verified using data to be collected in additional evaluation seasons (work in progress). The two most discriminant groups (associated with the first principal component) were the ones integrated by the solid aspects of bunch compactness; these correspond to berry-associated traits and the structure descriptors of the cluster, including the general dimensions of the bunch. Moreover, and remarkably, the dataset obtained by 3D scanning whole bunches showed a high degree of correlation with the ground measurements, suggesting that for every group of traits measured, at least one trait can be described using the very simple, fast, and precise 3D scan measurement process (see Figure 3 and Appendix A). The only exception is the group of traits associated with seed content, which escapes the possibility of being scanned (berries must be sliced and the seeds separated and cleaned to proceed with their measurement). Furthermore, the resolution of the scan is not appropriate for structures that are too small and thin, such as seeds or rachis. However, apart from these traits, we confirm the usefulness of the 3D scan process when using this population enriched for table grapes; this is very convenient for the evaluation of the large numbers of segregants managed in most breeding programs, which was already observed in several breeding populations of wine variety crossings [50], but this is the first report on table grapes, which have considerably larger bunches. Considering the high correlations between the ground and 3D measurements (almost all close to 95%), it would be interesting to evaluate the performance of the bunch compactness indexes compiled by Tello and Ibáñez [18] when using exclusively 3D scan-generated data, eventually defining new CIs better fitted to the data derived from this genetic background.

#### 3.3.2. Incorporation of Table Grapes in BC Studies

Table grapes were the main focus of this study, including wine as a comparison germplasm, which marks a difference regarding most previous works on grapevine bunch compactness. For example, Tello et al. [54] included only seven table grape varieties, with the remaining (over 100) varieties corresponding to wine or double-purpose genotypes. In our case, we considered a larger group of modern table grape varieties (most originated from different American breeding programs, and also a fraction of a nuclear collection was imported from INRA-France [55]) comprising old table and wine varieties coming from diverse geographical origins (Figure 5 and Appendix A). Additionally, a set of table grape lines provided by the Breeding Program of INIA was included in this study; in this way, we maximized the representation of the table grape genetic group.

Another reason to focus on table grape genetics is derived from a very practical aspect: the breeding activity in which markers are applicable is largely more intense in table grapes than in developing new wine varieties. Although this figure is slowly changing, a few wine varieties released during the last decades have found acceptance among growers and consumers, such as cv. Pinotage (South Africa), cv. Marselan (France), and the series of pioneering fungal disease-resistant cultivars released by the JKI-Geilweilerhof Institute in Germany [1,56]. On the contrary, most, if not all, of the planted table grape varieties have a very recent origin, comparable to other woody fruit crops. In fact, over the last decades, the table grape varietal turnover has been very dynamic, with dozens of new releases propagated and used by growers from around the world [57], highlighting the fact that from the last 60+ released varieties recorded in the VIVC database in the last two decades, only one was for wine, with the remainders belonging to the table grape group.

#### 3.3.3. Discussing the Role of the Enriched Terms Found in This Study

In the present work, the mining of candidate genes was restricted to a few over-represented biological functions. Specifically, we found genes related to glycan degradation, vesicular (organellar) transport, amino acid metabolism, and lipid metabolism, with sphingolipid and alpha-linolenic acid (ALA) metabolism being the most prevalent (Appendix A). This series of genes could be related to woody tissue such as rachis, and, as such, it depends on a complex combination of cell wall and membrane components, including phenylpropanoid and lipidic components. Of this series of genes/biological functions, ALA metabolism could perfectly fit with those biological activities. With respect to that, even when the record of studies on genes related to ALA metabolism in plants and their effects is far from abundant, it is noteworthy that this metabolite is a precursor to jasmonic acid (JA), which has been labeled as the stress response phytohormone that is also related to plant organ growth and other relevant cellular processes associated with crop resilience [58]. Specifically, the proposed roles for ALA in plants are quite diverse: the control of dormancy, pointed out in a woody species such as the European pear based on transcriptomic and metabolomic evidence [59]; participation in anther formation and male sterility, as evidenced in maize [60]; and a key role as a precursor to a number of metabolites, which are derived from a cascade of effects in diverse cellular processes specific to plants [61]. In the case of grapevines, in cv. Marselan, ALA has been linked to generating volatile organic compounds in response to shadowing, pinpointed from transcriptomic and metabolomic studies [62]. A role for ALA in the defense machinery against phytopathogens, such as the Chinese wild species *Vitis davidii* against Colletotrichum sp. [63], has also been proposed. The possible link between those biological activities in grapevines and the BC phenotype is, however, not evident and would require further studies. In fact, a more direct relation between ALA levels and BC phenotype has arisen from the findings that cellular turgor induced by mechanical compression increases the level of JA-Ile [64] and JA participates in the control of flowering [65]; the latter presumes that flower formation is connected to the fruit setting, and so in this way, it could determine, to some extent, berry number.

On the other hand, when using a less strict model (GLM), a considerable number of SNPs appeared to be significantly associated with various sub-traits of BC, such as berry polar diameter (or berry length). One of them was an SNP on chr 8 (position 8:2647579), which mapped close to Vitvi08g00144, a putative protein containing an LOB-20 domain, which has been found in transcription factors associated with susceptibility to Fusarium (yellowing) in Arabidopsis (and also through a JA signal transduction system) [66]. In general terms, LBD20 belongs to a family of transcription factors that have a domain called LATERAL ORGAN BOUNDARIES (LOB). The LOB domain is a conserved region located in the N-terminal portion of the protein [67], characterized by the presence of a C-block, a Gly-Ala-Ser (GAS) sequence, and a double-helix motif, such as a Leucine-zipper [68]. Interestingly, a highly significant signal associated with berry width (an SNP linked to the gene Vitvi11g000454) was recently identified using GWAS in a large collection of table grape segregants; it was located on chr 11 and was also related to plant response to biotic and abiotic stresses through JA signaling [45]. In this way, a possible route to future studies to understand BC in grapevine clusters might be found by exploring JA-associated pathways.

### 3.4. Limitations and Future Directions

A critical aspect of this work was population size. GWAS studies usually consider a larger number of genotypes than the figure considered here. However, in woody species such as grapevine, the numbers tend to be similar to this case [69].

A point to consider in future studies (for this or other traits) is the convenience of including (in the analyses) not just SNPs but also other polymorphisms, such as InDels or other sources of genomic diversity. However, most analytical packages have been designed to work with binary data, as most SNPs are. In our dataset, for instance, non-binary SNPs (tri-nucleotide variants) were present in a low proportion, increasing to around just 10% of the total filtered SNPs. Moreover, InDels are usually a minor fraction of structural variants; in this case, they accounted for close to one-third with respect to the prevalent SNPs, as was determined in the final group of multi-filtered markers. Another aspect to consider is the sanitary condition of the plants, as the presence of different phytopathogens can interfere with several determinants of capacity, especially for the solid component associated with berry number and size, or through the reduction of the general vigor of the plant and its organs. In this study, rigorous management was applied to control different pests, including fungi and insects. Even when the plants used in this work were tested and demonstrated the absence of problematic viruses, this is a dynamic scenario, and some infection levels could interfere with the performance of the vines.

This work explicitly excluded the use of gibberellin (represented by the synthetic GA_3_), the most important plant growth regulator (PGR) used in table grape production. Most modern varieties are seedless, with the seed being the main source of gibberellin. GA_3_ externally applied induces berry growth, with a direct incidence on bunch compactness (as reviewed by [18]). GA_3_ is also used for rachis elongation and flower thinning at different times and doses. For instance, previous work from our group related to the effect of GA_3_ on berry size led to the identification of sets of genes differentially expressed under the effect of this PGR [70]. Additionally, GA_3_ can affect post-harvest berry drop, altering the rigidity of the pedicels through the activation or repression of other sets of genes [71]. Thus, the impact of GA_3_ on cluster architecture would be an interesting area to study after this project, especially when also considering recent evidence of the effect of GA_3_ applications on rachis elongation being effective in its last portion and not in every part of this structure (such as the upper part of the central axis, the shoulders/arms, and the pedicel lengths, as it is generally assumed) [72]. Therefore, a separate gene expression/transcriptomic analysis of each of these rachis portions after GA_3_ application could provide useful information regarding the optimization of table grape vineyard productive management.

## 4. Materials and Methods

### 4.1. Plant Material

A collection of 116 grapevine genotypes was considered for this study. This set was composed of 24 wine, 36 table, and 26 mixed-usage cultivars, plus 30 table grape segregating lines derived from 5 biparental crossings (Reyes-Impellizzeri et al., in preparation). The list of varieties and additional information can be found in Appendix A. The true-to-typeness of the varieties was verified using the standard set of nine SSR markers, as suggested by OIV (https://www.oiv.int/node/3112, accessed on 1 April 2021), corresponding to OIV codes 801 to 809. The number of plants for each cultivar ranged from 3–6 individuals. In the case of the segregating lines, just one plant per genotype was available. The plants were grown in a field on their own roots, trained under a Gouyet training system, and subjected to standard management practices regarding pruning, fertilization, watering, and preventive pest control. The experimental orchard is located at (33°34′20″ S; 70°37′32″ W; 630 m.a.s.l.) (INIA La Platina Experimental Station. Santiago, Chile).

### 4.2. Sampling

As the main experimental unit for the phenotypic data, bunches were sampled at harvest, which corresponded to E-L 38 developmental stage according to a phenological scale modified by Coombe [73]; according to weekly measurements of total soluble solids, the samples averaged 16–17 °Brix and 20–22 °Brix for table and wine grapes, respectively.

For the genotyping-by-sequencing (GBS) method, young and not fully expanded leaves were collected and frozen in liquid nitrogen. Then, plant material was stored at −80 °C until further DNA extraction (Reyes-Impellizzeri et al., in preparation).

### 4.3. Phenotypic Data: 3D Scanning and Manual Measurements

Harvested bunches were put on a hook connected to a motorized device with a rotation period of 18 s. Then, the 3D data acquisition was performed using an Artec 3D spider scanner (Artec 3D, Senningerberg, Luxembourg), as described by Rist et al. [17]. After registering the structure of each bunch through 3D imaging, the following variables were measured through conventional phenotyping. Bunches were weighed on a standard laboratory balance. Then, total berries were detached from the rachis and counted. Next, 10 berries were randomly selected from the total, and the polar and equatorial diameters (OIV descriptors N° 220 and 221 for berry length and width, respectively) were measured using a digital caliper. Berry fresh weight was estimated by averaging these 10 berries on an analytical balance; berries were dissected to count the number of seeds, and the seed fresh weight was estimated in a similar fashion. After gauging the rachis, the number of lateral branches was counted, and the stalk length was measured along with the first ramification length, second ramification length, and the internode 1–2 length. As suggested by the OIV descriptor N° 238, 15 pedicels were randomly selected, and their length was measured using a digital caliper. Some of these measurements were considered as empirical evidence for comparison with the data extracted from the 3D scans.

Regarding the data extracted from the scanning, the data were exported and a point cloud-based archive was generated after processing using the 3D Bunchtool software v2.0 [17,36]. Then, berry detection was performed using the same software, and the following variables were extracted: position of each berry, diameter of each berry, berries per bunch, mean berry diameter, mean berry volume, total volume of the berries, convex hull volume, grape width, and grape height. A general scheme of the measured traits for both methods can be seen in Figure 1.

### 4.4. Genotyping-by-Sequencing and Variant Discovery

The samples were extracted using the DNease Plant Mini Kit (QIAGEN, Hilden, Germany) following the manufacturer’s instructions. The DNA aliquots were quantified and evaluated for purity using the Qubit dsDNA Broad Range assay from Invitrogen on a Qubit fluorometer (Invitrogen, Waltham, MA, USA). Subsequently, GBS libraries were prepared for all DNA samples using the restriction enzyme ApeK I, and the single pool generated from the libraries was sequenced on an Illumina NextSeq 550 HO flow cell at the University of Minnesota Genomics Center (Minneapolis, MN, USA) (Reyes-Impellizzeri et al., in preparation). The single-end reads (~150 bp) were trimmed using Trimmomatic with a Q20 filter. The trimmed reads were aligned to the grapevine reference genome PN40024 version 12X.2 with default parameters. Variant calling was performed using GATK (v4.1.4.0) [74]; “MarkDuplicatesSpark”, “HaplotypeCaller”, and “GenotypeGVCFs” commands were employed with default parameters to generate a variant call format (VCF) file [75]. Then, the variants were filtered using the “VariantFiltration” command to filter out those that met at least one of the following criteria: QD < 8.0, QUAL < 100.0, FS > 60.0, SOR > 3.0, DP < 3, DP > 30, or AD < 2. Freebayes was used to call variants jointly across all samples using the genotype qualities option and min-coverage 240 [76].

### 4.5. Genotype–Phenotype Association Studies

A total of 685,934 variants (a Phred quality score of >Q20) (Reyes-Impellizzeri et al., in preparation) were discovered. These variants were filtered by keeping only those with a minimum allele frequency of 0.05, a maximum missing data of 0.05, a minimum mean depth read of 10, and a maximum read depth of 1000, reducing the number of potentially informative variants to 114,515. The InDel variants were filtered out, and only biallelic SNPs were kept, yielding a total of 70,335 markers. These SNPs were imputed using Beagle v5.2 [77,78]. GEMMA software v0.98.5 was used to perform the genotype–phenotype association analysis on the samples, using a mixed linear model (MLM) [46]. The centered relatedness matrix was used to estimate the associated SNPs. Given the multiple hypothesis test error rate, the obtained *p*-values were corrected using the false discovery ratio correction method [47]. Bonferroni family-wise corrected values are also shown as a reference for some results.

### 4.6. In Silico Annotation of Closest Genes Regarding Markers with Significant Association

SNPeff software v5.2f [79] was used to annotate the genes around the called SNPs within a window of 5000 bp (2500 upstream and downstream) from such variants. The custom database was produced using version 3 of the annotated file for *V. vinifera* L. [44]. Significant SNPs that appeared in several traits are detailed with their respective descriptions based on the predicted annotations.

### 4.7. Enrichment Analysis of Gene Ontology Categories

The ShinyGO web application was used to perform the gene-set enrichment analysis [80]. A list of 92 SNPs present in both seasons from all the traits examined was used to test the over-representation of gene ontology (GO) categories. Most of the SNPs were present in several seasons (Appendix A). Still, only one entry per variant was kept, with the closest annotated gene in a window of 5000 bp (2500 bp upstream and 2500 downstream) on each entry being taken; thus, a list of 64 genes was the input for the enrichment analysis.

### 4.8. Statistical Analyses

The output data were processed using the R language and environment (R core team, 2024). Descriptive and inferential statistics were managed using the following packages: stats (R base), Adegenet [81] for multivariant analyses of SNP marker data, vcfR [82] for overall quality control of marker data and variant distribution, ggtree [83] for dendrogram representation, and ggplot2 [84] for most of the graphical output.

## 5. Conclusions

Automated phenotyping based on 3D scanning is suitable for datasets detailing table grapes. The correlation values for a collection of 116 genotypes composed of 24 wine, 66 table, and 26 double-purpose genotype backgrounds (comparing inferred values against ground measurements) revealed high correlation values.The analyses of these datasets collected over two consecutive seasons led to the identification of a large set of significant SNPs when using the GWAS approach. The most striking signals were associated with the number of berries per bunch (BB). Other signals for other BC-related traits were distributed on various chromosomes, with high variability among seasons. The list of genes closest to the significant SNPs revealed significant enrichment in discrete functions such as alpha-linolenic acid and glycan degradation, which poses new questions regarding the BC phenotype.Understanding the genetic bases in both genetic groups is highly relevant under climate change scenarios, which are driving the development of new table grapes and wine varieties.

## Figures and Tables

**Figure 1 plants-14-01308-f001:**
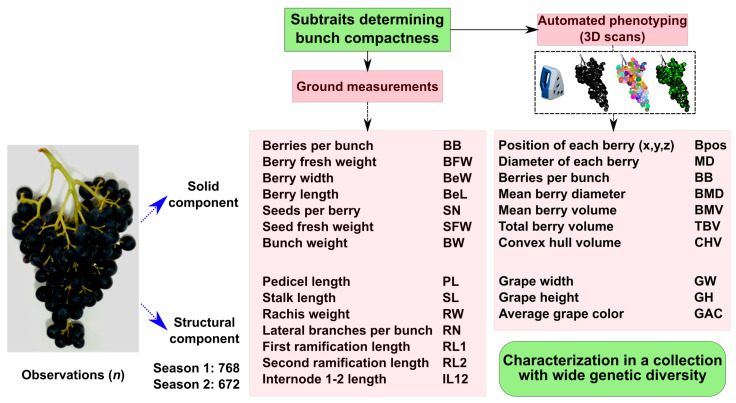
Methodological scheme for the characterization of traits determining bunch compactness. To capture the complexity of the bunch compactness (BC) phenotype, several attributes that determine the overall compactness degree in grapes were measured (discussed in [19]). Two approaches were taken to address BC: (1) ground measurements through traditional phenotyping and (2) 3D scanning methodology followed by automated phenotyping (described in [17]).

**Figure 2 plants-14-01308-f002:**
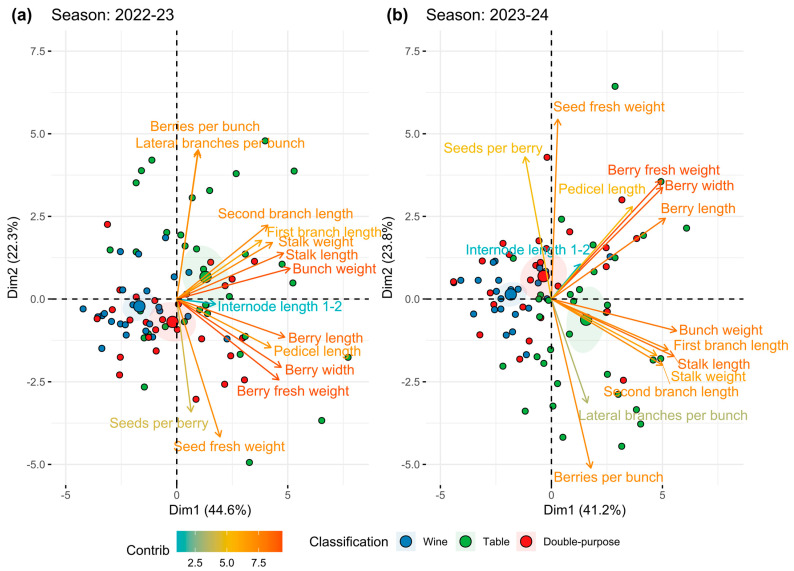
Multivariate analysis of quantitative traits describing bunch compactness suggests four groups associated with discrete morphological aspects. The ground measurement data were inspected through principal component analysis to reduce the complexity of the dataset and explore potential relationships among traits. Similar patterns resulted in two consecutive seasons, namely (**a**) 2022–2023 and (**b**) 2023–2024. Two groups are closely related to the first principal component: one is related to the solid aspect of bunch compactness, which is centered in the berry organ, and the other group is related to the structural aspect in terms of overall dimensions. The other two groups are more related to the second principal component, with higher variability among seasons; these groups are inversely correlated and contain fewer traits, with the number of berries/ramifications being opposed to the seed content. Each circle corresponds to a genotype, cataloged as indicated by color codes.

**Figure 3 plants-14-01308-f003:**
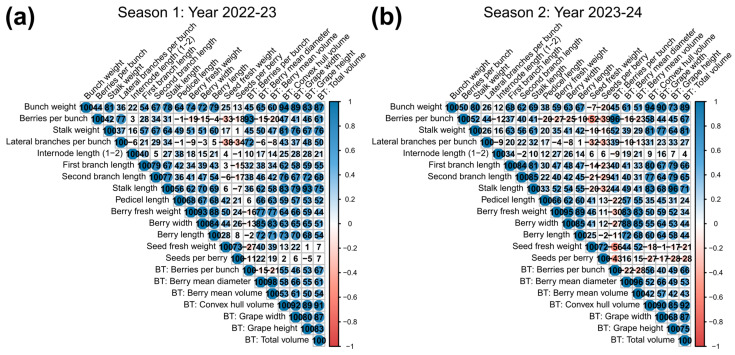
Exploring the correlation degree between traits inferred by 3D automated phenotyping versus ground measurements. All traits measured using manual methods (empirical evidence) versus those inferred by 3D automated phenotyping (labeled as ‘BT’, Bunchtool) were compared over two seasons, (**a**) 2022–2023 and (**b**) 2023–2024, resulting in the correlation matrix depicted here. Pearson correlation R^2^ is presented as a percentage and is the result of comparing the data collected from the 116 genotypes included in the study.

**Figure 4 plants-14-01308-f004:**
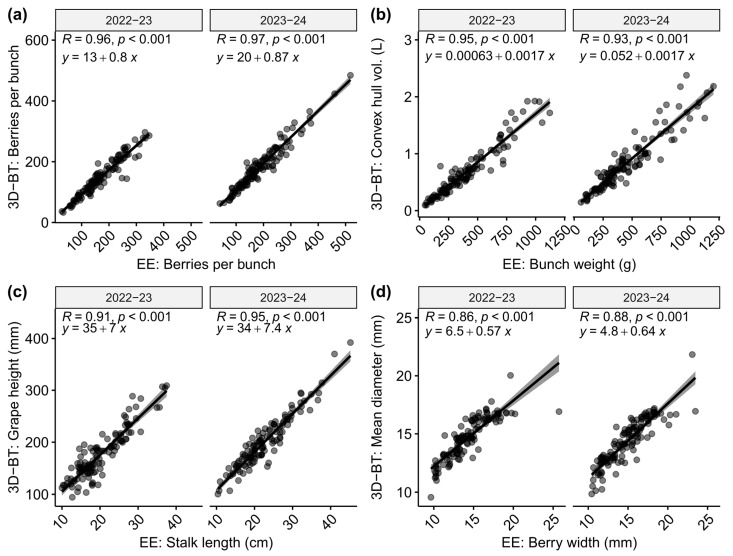
High correlation of empirical values versus inferred values from 3D automated phenotyping. Correlation analysis of data from the average values of 116 genotypes showed a strong association between comparable traits. (**a**) Example of direct comparison corresponding to the number of berries per bunch. (**b**) The convex hull volume showed a high correlation with bunch weight. (**c**) Bunch height showed a high correlation with measured stalk length, as expected. (**d**) The mean berry diameter for the fitted spherical shapes was compared against the berry width; the deviation was relatively mild, except for some data points from genotypes with elliptical bodies.

**Figure 5 plants-14-01308-f005:**
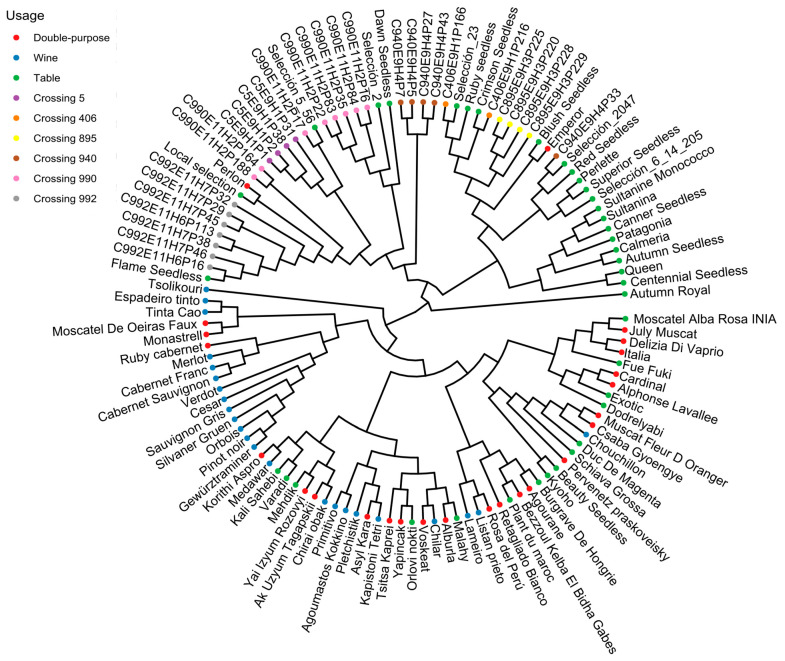
Genetic relationships among genotypes considered for the study of BC. The dendrogram is based on hierarchical clustering following the UPGMA method, using the Euclidean distances from a set of 70,335 informative SNPs discovered through a GBS assay. Regarding categories, the genotypes and advanced/selected lines were classified according to their reported usage, namely wine, table, or double-purpose. Nameless genotypes, which are seedlings from the table grape breeding program of INIA [45], are labeled according to the crossing or families to which they belong.

**Figure 6 plants-14-01308-f006:**
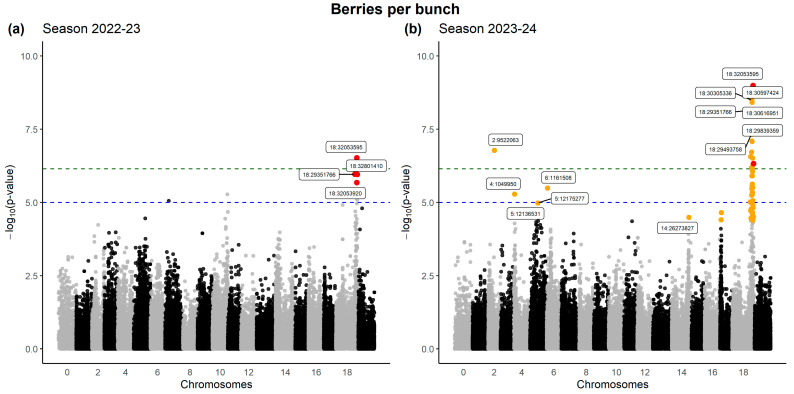
Identification of SNPs associated with berries per bunch (BB), a key trait determining bunch compactness. The values detected by Bunchtool software v2.0 were intersected with the genetic data provided by 70,335 biallelic markers, which were analyzed through linear mixed models under GEMMA [46]. *p*-values corrected using the false discovery ratio method are highlighted in orange. Additionally, common significant SNPs between seasons are highlighted in red. The green dotted line corresponds to the Bonferroni threshold (family-wise correction). The blue dotted line is a suggestive threshold for significance (1 × 10^−5^). Two consecutive seasons were evaluated: (**a**) Season 2022-23 and (**b**) Season 2023-24.

**Table 1 plants-14-01308-t001:** Potential effects of significant SNPs on their closest genes for two descriptors of bunch compactness. A list of SNPs that were significant for two variables describing BC is detailed here. False discovery ratio correction was applied to differentiate the variants that were truly significant. For each SNP, the affected gene is depicted along with a brief description based on sequence similarity. Most of the significant SNPs were found in the second season, as detailed in the list. For chr 18 during the second season, a total of 43 SNPs were filtered out, which spanned a region from 29351766:32801410. The full list can be accessed in Appendix A, along with the significant SNPs detected for the other traits considered in this study.

Trait	Season	Chr	Position	*p*-Value (Raw)	Ref	Alt	Annotated Effect	Candidate Gene	Description
BB	2022–2023	18	29,351,766	1.10 × 10^−6^	G	A	3_prime_UTR_variant	Vitvi18g02073	Adenylate kinase, chloroplast
BB	2022–2023	18	32,053,595	3.00 × 10^−7^	T	C	upstream_gene_variant	Vitvi18g02260	
BB	2022–2023	18	32,053,920	2.10 × 10^−6^	G	T	missense_variant	Vitvi18g03207	
BB	2022–2023	18	32,801,410	1.10 × 10^−6^	C	T	synonymous_variant	Vitvi18g02321	Aspartokinase–homoserine dehydrogenase
BB	2023–2024	2	9,522,063	1.70 × 10^−7^	C	T	intron_variant	Vitvi02g01526	NADP-malic enzyme
BB	2023–2024	4	1,049,950	5.20 × 10^−6^	G	T	upstream_gene_variant	Vitvi04g01789	
BB	2023–2024	5	12,136,531	1.00 × 10^−5^	G	C	intron_variant	Vitvi05g02021	Zinc finger (C3HC4-type ring finger)
BB	2023–2024	5	12,175,277	1.00 × 10^−5^	A	G	downstream_gene_variant	Vitvi05g00968	Zinc finger (C3HC4-type ring finger)
BB	2023–2024	6	1,161,508	3.20 × 10^−6^	C	G	upstream_gene_variant	Vitvi06g00101	
BB	2023–2024	14	26,273,827	3.20 × 10^−5^	G	A	missense_variant	Vitvi14g01604	F-box domain containing protein
BB	2023–2024	17	1,640,664	3.90 × 10^−5^	G	T	upstream_gene_variant	Vitvi17g01358	Homeobox protein knotted-1-like 1 (KNAT1)
BB	2023–2024	17	2,126,568	2.20 × 10^−5^	A	G	upstream_gene_variant	Vitvi17g00197	POT1 (protection of telomeres 1)
BB	2023–2024	18	29,351,766	7.10 × 10^−9^	G	A	3_prime_UTR_variant	Vitvi18g02073	Adenylate kinase, chloroplast
BB	2023–2024	18	32,053,595	1.00 × 10^−9^	T	C	upstream_gene_variant	Vitvi18g02260	
BB	2023–2024	18	32,053,920	9.00 × 10^−11^	G	T	missense_variant	Vitvi18g03207	
BB	2023–2024	18	32,801,410	4.70 × 10^−7^	C	T	synonymous_variant	Vitvi18g02321	Aspartokinase–homoserine dehydrogenase
CI-03	2023–2024	2	9,522,063	1.90 × 10^−6^	C	T	intron_variant	Vitvi02g01526	NADP-malic enzyme
CI-03	2023–2024	14	25,584,904	1.70 × 10^−6^	T	C	synonymous_variant	Vitvi14g01530	Nodulin
CI-03	2023–2024	14	26,067,440	9.10 × 10^−6^	T	C	upstream_gene_variant	Vitvi14g02428	Transcription termination factor mitochondrial mTERF
CI-03	2023–2024	14	26,067,815	8.70 × 10^−6^	A	T	upstream_gene_variant	Vitvi14g02428	Transcription termination factor mitochondrial mTERF
CI-03	2023–2024	14	26,097,728	1.10 × 10^−6^	T	C	3_prime_UTR_variant	Vitvi14g01585	Phosducin
CI-03	2023–2024	14	26,097,791	2.30 × 10^−6^	A	T	3_prime_UTR_variant	Vitvi14g01585	Phosducin
CI-03	2023–2024	18	29,351,766	1.60 × 10^−6^	G	A	3_prime_UTR_variant	Vitvi18g02073	Adenylate kinase, chloroplast
CI-03	2023–2024	18	32,053,595	1.40 × 10^−6^	T	C	upstream_gene_variant	Vitvi18g02260	
CI-03	2023–2024	18	32,053,920	3.40 × 10^−7^	G	T	missense_variant	Vitvi18g03207	

**Table 2 plants-14-01308-t002:** Genes affected by the significant SNPs detected across all evaluated traits. A list of the genes affected by more than one instance (the same SNP appearing in multiple traits/seasons or a different SNP affecting the same gene) was compiled, as detailed in the third and fourth columns.

Gene ID	Description	No. of Hits	Traits on Which Associated SNP Was Detected
Vitvi11g00545	Chloride channel protein CLC	6	S1: CI_18 (3x); S2: CI_18 (3x)
Vitvi18g02073	Adenylate kinase, chloroplast	5	S1: BT_BB, RN; S2: BB, BT_BB, CI_03
Vitvi18g02242		2	S1: CI_17; S2: BT_BB
Vitvi18g02260		4	S1: BT_BB; S2: BB, BT_BB, CI_03
Vitvi18g03207		4	S1: BT_BB; S2: BB, BT_BB, CI_03
Vitvi18g02321	Aspartokinase–homoserine dehydrogenase	4	S1: BT_BB; S2: BB, BT_BB (2x)
Vitvi02g00318	Tyrosyl-tRNA synthetase	2	S1: CI_19 (2x)
Vitvi02g00387	Heat shock transcription factor B2B	2	S1: CI_17, CI_19
Vitvi02g00399	SEU1 protein	4	S1: CI_17, CI_19 (3x)
Vitvi02g00417	Copper amine oxidase	3	S1: CI_17, CI_18, CI_19
Vitvi02g00607	Unknown protein	2	S1: CI_17, CI_18
Vitvi02g01526	NADP-malic enzyme	3	S2: BB, BT_BB, CI_03
Vitvi03g00829		2	S1: CI_08, RW
Vitvi04g01789		2	S2: BB, BT_BB
Vitvi04g00282	Lachrymatory factor synthase	3	S1: CI_17, CI_18, CI_19
Vitvi04g00374	DnaJ homolog, subfamily B, member 4	2	S1: CI_17 (2x)
Vitvi04g00375	Fertility restorer homologue A	2	S1: CI_17 (2x)
Vitvi04g00732	Ubiquitin-specific protease 15	2	S2: BT_BMV (2x)
Vitvi05g02021	Zinc finger (C3HC4-type ring finger)	2	S2: BB, BT_BB
Vitvi05g00968	Zinc finger (C3HC4-type ring finger)	2	S2: BB, BT_BB
Vitvi05g00990	DNA primase large subunit	3	S2: BB, BB, BB
Vitvi06g00101		2	S2: BB, BT_BB
Vitvi06g00538	Chlororespiratory reduction 2 (CRR2)	2	S1: CI_17, CI_18
Vitvi07g00654	Ankyrin	2	S1: BFW, PL
Vitvi07g01536	Major facilitator superfamily protein (MFS) Spinster	3	S1: CI_17, CI_18, CI_19
Vitvi07g01999	U3 small nucleolar ribonucleoprotein protein IMP4	2	S2: CI_12, CI_16
Vitvi08g02237		2	S1: CI_17, CI_19
Vitvi09g00430	EMB2753	3	S1: CI_17 (2x), CI_18
Vitvi10g00060		4	S2: CI_17 (2x), CI_18 (2x)
Vitvi10g01508	F-box/LRR-repeat protein 2	3	S2: RL2 (3x)
Vitvi11g00440	Zinc finger (CCCH-type) family protein	4	S1: CI_17 (2x), CI_18 (2x)
Vitvi11g00475	EIN3-binding F-box protein 1	4	S1: CI_17 (2x), CI_18 (2x)
Vitvi11g01443		4	S1: CI_17 (2x), CI_18, CI_19
Vitvi11g00487	Glucan endo-1,3-beta-glucosidase 7 precursor	2	S1: CI_17, CI_18
Vitvi11g00512	Unknown protein	2	S1: CI_17, CI_18
Vitvi11g00517	CCR4-NOT transcription complex subunit 10	4	S1: CI_17 (2x), CI_18 (2x)
Vitvi11g00523	Clathrin assembly protein 10	2	S1: CI_17, CI_18
Vitvi11g00539	HHP1 (heptahelical protein 1)	2	S1: CI_17, CI_18
Vitvi11g00542	Sucrose-phosphate synthase	4	S1: CI_17 (2x), CI_18 (2x)
Vitvi11g00543	Nodulation receptor kinase	7	S1: CI_17 (3x), CI_18 (4x)
Vitvi11g00549	IMP dehydrogenase/GMP reductase	4	S1: CI_17 (2x), CI_18 (2x)
Vitvi12g00056	1,3-beta-glucan synthase	2	S2: BB, BB
Vitvi12g00124	S-receptor kinase	2	S2: CI_12, CI_16
Vitvi12g02121	U2(RNU2) small nuclear RNA auxiliary factor 1-like 2	2	S2: CI_18, CI_19
Vitvi13g01903		3	S1: CI_17, CI_18, CI_19
Vitvi13g00145	Myosin heavy chain	3	S1: CI_17, CI_18, CI_19
Vitvi13g01924	Cytochrome c oxidase subunit VIb	2	S1: CI_18, CI_19
Vitvi13g00779	ABC transporter G member 22	2	S1: CI_17, CI_19
Vitvi14g01171		2	S1: CI_17 (2x)
Vitvi14g01232	Nuclear matrix constituent protein 1	4	S1: CI_17 (2x), CI_18, CI_19
Vitvi14g01448	Chalcone synthase (VviCHS1)	2	S1: CI_17, CI_19
Vitvi14g02428	Transcription termination factor mitochondrial mTERF	2	S2: CI_03 (2x)
Vitvi14g01585	Phosducin	4	S2: CI_03 (4x)
Vitvi14g01604	F-box domain containing protein	2	S2: BB, BT_BB
Vitvi15g00571	Palmitoyl-protein thioesterase 1 precursor	3	S1: CI_17, CI_18, CI_19
Vitvi15g00585		2	S1: CI_18, CI_19
Vitvi16g00899	Subtilisin protease C1	2	S1: CI_17 (2x)
Vitvi16g01191		3	S2: BT_BMV (3x)
Vitvi17g00284	Kinesin motor HIK (HINKEL)	2	S1: CI_17, CI_19
Vitvi17g00288	Receptor serine/threonine kinase PR5K	2	S1: CI_17, CI_19
Vitvi17g00289	Wall-associated kinase 4	5	S1: CI_17 (2x), CI_18, CI_19 (2x)
Vitvi17g00299	EMB2454 (embryo defective 2454)	3	S1: CI_17, CI_18, CI_19
Vitvi17g00321	Kinesin motor protein-related	3	S1: CI_17, CI_18, CI_19
Vitvi18g00533	CTR1-like protein kinase	2	S2: CI_18 (2x)
Vitvi18g01662	Sensitive to proton rhizotoxicity 1	4	S1: CI_17 (2x), CI_18 (2x)
Vitvi18g01948		2	S2: BT_BB (2x)
Vitvi18g01981	Proton-dependent oligopeptide transport (POT) family protein	2	S2: BB, BT_BB
Vitvi18g02081	Protein translocon component Tic40, chloroplast	3	S2: BB, BT_BB, CI_03
Vitvi18g02103	Taxane 10-beta-hydroxylase	5	S2: BB (2x), BT_BB (2x), CI_03
Vitvi18g03151		2	S2: BT_BB (2x),
Vitvi18g02121	Nematode resistance protein	3	S2: BB, BT_BB, CI_03
Vitvi18g02122		2	S2: BB, BT_BB
Vitvi18g02128	Glyoxylate reductase	4	S2: BB (2x), BT_BB (2x)
Vitvi18g02133	MADS-box protein SEEDSTICK (VviAG3)	5	S2: BB, BT_BB (2x), CI_03, ED
Vitvi18g03155	GORK (gated outwardly rectifying K+ channel)	2	S2: BB, BT_BB
Vitvi18g02139	12-oxophytodienoate reductase 2	2	S2: BB, BT_BB
Vitvi18g03160	12-oxophytodienoate reductase 1	7	S2: BB (3x), BT_BB (3x), CI_03
Vitvi18g02142	12-oxophytodienoate reductase 1	3	S2: BB, BT_BB, BT_CHV
Vitvi18g02159	R protein disease resistance protein	2	S2: BB, BT_BB
Vitvi18g02171	Aldehyde oxidase 1	2	S2: BB, BT_BB
Vitvi18g02180	Disease resistance protein (TIR-NBS-LRR class)	4	S2: BB, BT_BB (3x)
Vitvi18g02184		4	S2: BB (2x), BT_BB (2x)
Vitvi18g02185	Aminoacyl-tRNA synthetase, class Ia	2	S2: BB, BT_BB
Vitvi18g02186		4	S2: BB (2x), BT_BB (2x)
Vitvi18g02215		2	S2: BB, BT_BB
Vitvi18g02225		2	S2: BB, BT_BB
Vitvi18g02248		2	S2: BB, BT_BB
Vitvi18g02267	VviTPS89	2	S2: BB, BT_BB
Vitvi18g02462	VviTPS81	2	S2: BB, BT_BB
Vitvi18g02337	R protein L6	2	S1: CI_17, CI_18
Vitvi19g01412	Agenet domain-containing protein	4	S1: CI_14, CI_17, CI_18, CI_19

## Data Availability

Any data supporting our results will be shared upon request to interested researchers, following the FAIR principles.

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
