# Peer review of "Characterization of Bunch Compactness in a Diverse Collection of Vitis vinifera L. Genotypes Enriched in Table Grape Cultivars Reveals New Candidate Genes Associated with Berry Number"

_plants, 2025, doi:10.3390/plants14091308_

Round 1
Reviewer 1 Report
Comments and Suggestions for Authors
Please check the attachment.

Author Response
Reviewer 1: Major Revisions
- Many unusual words used in the scientific writing make the manuscript not easy to understand. Please polish it again.
Resp: We have revised the manuscript with special attention on the use of unusual words and sentences, trying to correct it as much as possible. In addition, we have used the MDPI English edition service, which has substantially improved the manuscript understanding. As a group, we hope this aspect is now properly addressed.
- P5, logical problems in Figure 2 and context. This part should be moved to P9 and integrated to 2.3 Genetic diversity and structure of the collection of samples. Logically, 2.1 and 2.2 are the analysis of phenotyping and 2.3 for genotype.
Resp: We fully agree with the reviewer’s rationale and consequently made all the suggested changes: Thus, the previous Figure 2 is the current Figure 5 in the revised version, moving it from page 5 to page 9. The other Figures were renumbered correspondingly in order of appearance.
- Figure 3 shows that most of the double-purpose genotypes overlap with Wine genotypes, please analyze the possible reasons in Discussion.
Resp: We appreciate the reviewer’s assessment. Consequently, we have included a brief analysis of this observation; in the current revised version of the ms, it is on page 15, 2nd paragraph of section 3.2.1 of the Discussion. We noticed such pattern of clustering in both Figure 2 (principal component analysis of phenotype data) and partially in Figure 5 (hierarchical clustering based on genetic data). Briefly, we discuss that this closer association of double-purpose varieties to wine varieties is because such categorization is based on the registered uses of each genotype which could not properly reflect the most prominent purpose in each case: e.g. in many cases an old wine variety (such as Listán Prieto) has been also traditionally used for fresh consumption, and consequently has been registered as double-purpose variety even when the latter use is less frequent. Both the BC-related phenotypic data and the genetic clustering evidenced in this work (the latter based on over 70,000 SNPs covering the whole grapevine genome) could eventually be used to reclassify some of the varieties included in this work as belonging or being closer to the wine or table grape groups.
- Comparing the Season 2022-23 and 2023-24, there are significant differences such as Figure 3 a and b, Figure 5, in two seasons, please further discuss the reasons.
Resp: We appreciate the reviewer’s comment addressing the existent variability among seasons, which can be evidenced in Figures 2 a/b and 4 in the current version of the ms. In the case of Figure 2 (previously Fig. 3), both datasets produce apparently very different patterns by seasons, however such differences can be explained due to a problem of perspective or orientation of the Y-axis, which is inverted when comparing (a) and (b). In other words, the relationship of the different genotypes (dots) and traits (line arrows) is conserved across seasons, even when the overall graph is inverted in one year respect of the other. We are aware that multivariate analysis output is dimensionless but observing the inversion in the Y-axis could be also indicative of the variability of traits related to the second component, and we wished to represent such behavior. In the case of Figure 4 (previously Fig. 5), the inter-season differences in the four graphs can be appreciated by the overall differences in scales (maximum and minimum values) which are more evident in the berries per bunch (a) and grape height (c) traits. We present both R2 values and linear equation, as indicative of the confidences of both methods and inherent variability of values in each trait by season, respectively. We observed close (and high) R2 values among seasons for each trait and in some cases high variability of the intercept and slope which is partially explained by a higher variability related to certain traits such as berries per bunch (a) and a lower variability in certain traits such as berry dimensions (d).
Rev. 1: Minor Essential Revisions
- P2, Line 60, “cépages” should be in English instead of French.
Resp: It was changed as indicated, thanks for the point.
- Please adjust the font size in Figures 1, 3 and 5 to make the font be coordinated with the context.
Resp: The Font size was adjusted in the indicated figures, making them better coordinated with the rest of the corresponding images. Some other minor adjustments were made to improve the Figures readability avoiding the overlap of labels in Figure 2 and modifying the scale of graph (d) in the current Figure 4 for better visualization (mm3 to Liters).
- Full form should be provided at the first abbreviations such as GBS in P4, Line 154, replace the full form in P5, Line 172, and deleted the full form in P9, Line 281. Please revise other abbreviations as well.
Resp: We have modified the text following reviewer’s indication, checking other similar cases too. Thanks
- P15, Line 385, change ‘3.2. ’ to ‘3.2’.
Resp: Done
- P17, Line 503, Fig. 2 should be Figure 2.
Resp: Thanks for noting it! Changed.
- P18, the first sentence of second and third paragraph is repeated, please revise them.
Resp: We have corrected this mistake, appreciate the comment!
- P18, Line 562, the expression ‘alpha-linolenic’ is inconsistent with before. The same professional word in the context should be in one word.
Resp: We have corrected this inconsistency revising the whole text. Thanks.
- P19, Line 611, what’s ca.10%?
Resp: ca. is the Latin abbreviation for circa, meaning “approximately”. It was changed, following the context of the sentence
- Inconsistency in Reference such as the issue name, there are three formats in the issue name such as abbreviation with dot of Front. Plant Sci., full form of Journal of Experimental Botany, and abbreviation without dot of BMC Plant Biol. Please uniform in one and standard format of Pl
Resp: Thanks for the point! We consequently uniformed the style of the references.
Reviewer 2 Report
Comments and Suggestions for Authors The document explores research centered on grapevine breeding, with a particular emphasis on traits influencing bunch compactness (BC) and their underlying genetic factors. Major insights from the study include: 1. Genetic Variation: The study utilizes a genetically diverse set of primarily table grape varieties, revealing significant genetic distinctions through genotyping-by-sequencing techniques.2. Bunch Architecture: Researchers dissect the components that shape bunch compactness, underscoring their relevance to both disease resistance and fruit quality. The complexity introduced by the polygenic nature of these traits is also addressed.
3. High-Throughput Phenotyping: The validation of an automated phenotyping system demonstrates its effectiveness in evaluating BC traits, enabling consistent and scalable data collection.
4. Genomic Associations: Newly identified single-nucleotide polymorphisms (SNPs) linked to BC-associated traits are presented, highlighting the benefit of using test populations that reflect the target genetic backgrounds.
5. Data Interpretation: A range of statistical tools was applied to analyze trait data, offering deeper insights that support grape breeding initiatives.
The findings have broader relevance for enhancing grapevine cultivars in response to evolving climate conditions and agricultural demands. The authors also note constraints, such as limited sample size and environmental variability, and propose future research to further investigate the genetic mechanisms involved. ---
The manuscript is well-written, but it should be improved slightly before publication according to the following recommendations:
Minor remarks
- The first-person plural should be avoided in the manuscript, while the third-person is only acceptable for scientific papers. Please, modify all sentences according to these recommendations.
Major remarks
- The Introduction should be shortened, as it is currently too long.
- References should be excluded from the Conclusion section. Only the main conclusions derived from the analysis should be presented in this section.
- Avoid grouping references. Each reference should be discussed individually.
Author Response
Minor remarks
- The first-person plural should be avoided in the manuscript, while the third-person is only acceptable for scientific papers. Please, modify all sentences according to these recommendations.
Resp: We have followed this very important reviewer’s suggestion, revising the whole text. It is most appreciated. A list of modifications and adjustments is at the bottom of this response, which are additional to the changes marked in the text of the current revised version of the ms, mostly corresponding to English edition of the text.
Major remarks
- The Introduction should be shortened, as it is currently too long.
Resp: We have shortened the Introduction. Most redundant sentences have been discarded, taking care of not altering the general meaning of each paragraph. A list of these changes is summarized in the following lines.
- References should be excluded from the Conclusion section. Only the main conclusions derived from the analysis should be presented in this section.
Resp: We have discarded the references from the Conclusions section, accordingly.
- Avoid grouping references. Each reference should be discussed individually.
Resp: Following the comment by the reviewer, we have modified the text individualizing the references in every sentence where they were grouped to support different specific cases
Note: List of changes introduced to optimize the text
Abstract:
- Edited on lines 19, 21, 23, 24, 28, 32-36 to avoid use of first-person plural and improve overall legibility
Introduction:
- L43-45: Edited to improve conciseness
- L49: Grouped references 3 to 6 were cited to their corresponding sentence
- L53: References 9 and 10 individualized to their corresponding sentence
- L61: Replaced “cépages” by “cultivars”
- L64-66: Deleted to reduce redundancy in the Introduction section
- L70-71: Deleted to improve conciseness
- L84-85: Deleted to improve conciseness
- L87-89: Deleted to improve conciseness
- L90-91: Deleted to improve conciseness
- L101-103: Deleted to improve conciseness
- L103-111: Edited to improve legibility
- L112-114: Deleted to improve conciseness
- L115-123: Edited paragraph to improve conciseness and legibility
- L147-153: Deleted to improve conciseness
- L162: Changed abbreviated terms
Results:
- L200: Edited to avoid use of first-person plural
- L276-279: Deleted to improve conciseness
- L326-330: Edited to avoid use of first-person plural
- L383-389: Edited to avoid use of first-person plural
- Changed “figure 2” from page 5 to page 9. Now corresponds to figure 5
Discussion:
- L402: Deleted to improve conciseness
- L412-416: Edited to avoid use of first-person plural
- L437: Edited to avoid use of first-person plural
- L477: Edited to avoid use of first-person plural
- L481: Edited to improve conciseness
- L508: Edited to avoid use of first-person plural
- L555: Edited to avoid use of first-person plural
- L565-570: Deleted redundant paragraph
- L583-584: Deleted to improve conciseness
Material and methods:
- L692-696: Modified to improve conciseness
- L753-755: Edited to avoid use of first-person plural
Conclusions:
- Modified to improve conciseness and legibility. Cited references were deleted
References:
- The references were standarized according to the Zotero reference manager “Multidisciplinary Digital Publishing Institute” style. In some cases, it required a hand-made correction, in order to uniform the style.
Other changes:
Several changes to figure captions and other minor changes to improve conciseness of the text, phrasing and others.